# Towards Creation of Ceramic-Based Low Permeability Reference Standards

**DOI:** 10.3390/ma12233886

**Published:** 2019-11-25

**Authors:** Svyatoslav Chugunov, Andrey Kazak, Mohammed Amro, Carsten Freese, Iskander Akhatov

**Affiliations:** 1Skolkovo Institute of Science and Technology (Skoltech), Moscow 121205, Russia; A.Kazak@skoltech.ru; 2Institute of Drilling and Fluid Mining, Technical University of Bergakademie Freiberg (TU BAF), 09599 Freiberg, Germany; Mohd-Musa-Mohd.Amro@tbt.tu-freiberg.de (M.A.); freese@mailserver.hrz.tu-freiberg.de (C.F.)

**Keywords:** Al_2_O_3_, 3D-printing, stereolithography, sintering, porosity, permeability

## Abstract

Low-permeable materials, either artificial or natural, are essential components of the current technological development. Their production or processing requires a comprehensive characterization method based on confident reference standards. Permeability standards for values below 10^−15^ m^2^ are lacking. In this study, we explored the feasibility of using the ceramic sintering process to reach low, but measurable values of gas permeability in Al_2_O_3_ samples, as one of the potential materials for reference standards. The studied samples were produced with a ceramic 3D printer, which enables the manufacturing of low-permeable samples having complex geometrical arrangements. A series of preliminary laboratory testing indicated the available gas permeability range from 2.4 × 10^−15^ m^2^ for the pre-sintered samples to 1.8 × 10^−21^ m^2^ for the sintered samples. The verification of the permeability reduction was carried out using a unique unsteady state fast and accurate measurement method. The results confirm the possibility of developing a technology for materials manufacturing with low porosity and permeability. Such materials open many areas for application, including manufacturing of ceramics with controlled transport properties and low-permeability standards for calibrating laboratory equipment for geosciences, construction industries, biomedical, and other relevant fields.

## 1. Introduction

Permeable and porous materials are crucial components of modern technological development, either natural or artificial; they play a significant role in scientific and industrial applications. Low-permeable materials are encountered on periodic basis in geological sciences when studying rock of tight reservoirs [1,2,3], cap rocks for CO_2_ sequestration [4,5,6,7], and underground storage for nuclear waste [8], crystalline, volcanic, metamorphic rocks, and salt deposits in mining industry [9,10]; in material sciences when developing ceramic and polymeric membranes [11,12,13,14,15], specialty glass and concrete [16,17,18,19], synthetic porous materials for catalysts and gas barrier applications, fuel cell membranes [20]; in biomedical field as contact lens materials and medical tools [21,22,23]. Despite the diverse implementation of low-permeable materials in science and industry, there is a limited number of laboratory characterization techniques that help to delve into material properties in terms of pore-space characteristics and permeability values.

Four laboratory methods are frequently utilized to assess the low-permeability of porous materials: the standard steady-state method, the transient technique, a direct application of Darcy’s law, and the pressure-pulse oscillation technique [24]. The typical fluids used for the assessment are water, air, nitrogen, argon, and helium. While such characterization is a well-established process for materials featuring permeability in milli-Darcy range and above, for low-permeable materials (below 10^−15^ m^2^), it is still a challenge [25,26]. A worldwide low-permeability testing cross-laboratory initiative [24] conducted at 24 universities and industrial laboratories with Grimsel Granodiorite samples resulted in a large discrepancy in the measured values even for the same testing method applied across different laboratories. This effort serves as a good indicator of the necessity of reference standard samples for calibrating laboratory equipment. The need for the reference standard samples is especially critical for permeability measurements below 10^−16^ m^2^ [27].

The need for reference permeability standards has been widely acknowledged in geosciences. The petrophysicists use permeability standards in standard and special core analysis, since the beginning of laboratory characterization of rock samples. The reference standards are routinely employed for calibrating permeability measurement equipment in the laboratories. In response to the growing demand in permeability standards, several manufacturing methods were proposed by industries to produce such items. The typical permeability range of the commercially produced permeability standards is 5 × 10^−16^–10^−13^ m^2^. For example, some existing cylindrically shaped reference standard items (Ø30 × 30 mm to Ø8 × 4–8 cm in size) were made of ceramic feedstock (85 vol.% of Al_2_O_3_ and 15 vol.% of titanium), pressed into cylindrical pieces, and sintered (Figure 1a,b). These samples demonstrate the relative stability of properties during measurements of open porosity (available range: 4.25–29.5%) and gas permeability (available range 2–965 × 10^−15^ m^2^), but low control of the properties during manufacturing.

The research group of S. Sinha [28] developed reference standards for permeability measurements in the range of 10^−20^–10^−17^ m^2^ (Figure 1c). These standards represent cylindrical items made of stainless steel, bored along the cylinder axis, and complemented with a glass micro-capillary, embedded into the bored annuli with the help of an impermeable epoxy adhesive. The permeability was calculated using the Hagen-Poiseuille equation for laminar flow in a circular conduit and verified via steady-state measurements with compressed air [29]. The permeability of the manufactured items was tested with the steady-state and the pulse-decay approach and compared to the results of the GRI (Gas Research Institute) testing technique [30]. The idea of creating the capillary-based permeability standards lead to the development of specialized kits for calibrating industrial and research nano-permeameters (Figure 2). The kits contain several Ø30 mm metal cylinders with axial inserts for permeability range of ~10^−20^–10^−19^ m^2^.

The practical application of the capillary-based permeability standards over the years indicated a high sensitivity of the items to the elements. The presence of any contaminant (oil, dust, chips) spoils the micro-channel and leads to flawed measurements. The single-channel arrangement makes cleaning of the items difficult, sometimes impossible, because of their negative response to various chemicals and high temperatures. Besides, the items are affected by a premature capillary condensation of ambient air humidity. These downsides hampered the manufacturing of the capillary-based reference standards at the industry-wide scale and limited their application in the laboratories.

An alternative technology for reference standard manufacturing is on-demand, to produce items targeting low-permeability range, not sensitive to elements, protected from contamination, easily cleanable, and inert to many liquids and gases, including petroleum derivatives. Ceramic materials seem to be good candidates because of their excellent physical-chemical properties. Even though examples of ceramic applications for reference standard manufacturing do exist, most ceramic units are manufactured “as is.” There is no technological method applied for qualitative prediction and control over the pore-space characteristics of the resultant item. The lack of data in the scientific literature on the processing of ceramic-based materials concerning the control over their pore-space characteristics and permeability in the nano- and micro-range indicates the necessity of research in this area. As a result, the establishment of a manufacturing practice to produce precise permeability reference standards is possible.

In this work, we present the preliminary results of laboratory measurements of porosity and permeability characteristics of debinded and sintered Al_2_O_3_ ceramics. We evaluate the efficiency of the sintering technique for material permeability modification and evaluate the available permeability range for Al_2_O_3_ ceramics. The two objectives pursued in this study are: (1) Test the porosity and permeability range provided with a broadly available technical ceramics, processed with a standard manufacturing procedure, and (2) test special low-permeability measurement equipment with non-standard ceramic samples.

## 2. Materials and Methods

### 2.1. Sample Manufacturing

For testing the technologically achievable permeability range of broadly used Al_2_O_3_ ceramics, a series of Ø 30 mm × 5 mm cylindrical ceramic samples were manufactured at Skoltech Additive Manufacturing Laboratory, with the help of stereolithography-based (SLA) 3D printing technology. The manufacturing of the samples was done with an SLA 3D printer Ceramaker 900 (3DCeram, Limoges, France) (Figure 3). The printer’s manufacturer provided the Al_2_O_3_ ceramic material. The material has a form of a highly viscous paste composed of ~65 vol.% ceramic powder and ~35 vol.% organic binder. The size of ceramic particles was measured via image analysis of SEM images. The traditionally used laser diffractometry technique does not correctly capture particle range for this material.

The 3D printing approach helps to assess the porosity and permeability range achievable via the SLA technology. The SLA technology allows the manufacturing of samples having complex geometric shapes for the future studies of complex porous and permeable items. The principles of the SLA technology are the following: A viscous UV-polymerizable ceramic paste loaded to a 3D printer is spread over a building platform into a 25–50 μm thick layer—the thickness of the layer depends on material type and size of the ceramic particles in the paste; a UV laser selectively polymerizes the required contours in the paste layer with accuracy of 50 um; the building platform descends by the layer thickness and the process repeats. Multiple successive layer-by-layer polymerizations build a 3D body. The 3D printed object is being cleaned off the unpolymerized paste and thermally processed to get the final article.

The thickness of the cylinders, manufactured with SLA technology, was limited to 5 mm because of the technological limitations of the standard SLA technique. After 3D printing the samples were cleaned with CeraClean (3DCeram, Limoges, France) solvent, applied with the compressed air. A “standard” debinding procedure provided by 3DCeram company was used for debinding of the samples. The organic binder that holds the green part together is thermally removed in the regular atmosphere during one of the post-processing steps. The organic components of the binder, when exposed to temperatures up to 600 °C decompose into gas molecules, exit the greenbody volume, and get evacuated by the ventilation system of a furnace. If the binder rapidly decomposes in material’s bulk, when the boundary regions are not yet clean of binder, the manufactured item fractures because of internal pressure build-up. It is recommended to use sample thickness smaller than 3 mm to avoid fracturing. In our practice, samples of 5 mm thick, having flat geometry, were successfully processed with no fractures observed.

The printed samples were gradually debinded, reaching temperatures around 600 °C, using Kittec CLL-15 debinding oven (Kittec, Rosenheim, Germany). The debinded samples had deficient mechanical strength; therefore, the samples were presintered at 1150 °C, following the printer manufacturer’s suggested procedure. The pre-sintered samples experienced minimal (less than 1%) reduction in geometry and acquired sufficient mechanical strength for further handling and processing with laboratory equipment, including the application of gas pressure.

The presintered samples have the highest porosity and permeability values achievable via the SLA approach. Further thermal processing decreases both characteristics. The porosity and permeability of the pre-sintered samples were measured at Skoltech Hydrocarbon Recovery laboratory. The presintered cylinders were assembled into a 30 mm thick vertical stack. A rubber gasket insulated the side of a stack, which is routine for permeability measurement equipment. The porosity and permeability of the presintered stack were measured with standard petrophysical laboratory equipment.

After the measurements, the stack was disassembled, and the samples were sintered at 1700 °C for 1.5 h, using ThermConcept HTL 20/17 sintering kiln (ThermConcept, Bremen, Germany). A standard thermal processing recipe, provided by the printer’s manufacturer, was followed. During sintering, the samples shrunk by 13–15% in linear dimensions. The porosity and permeability of the samples were significantly reduced during sintering. A unique experimental setup developed at TU BAF for accurate low-permeability measurements was used to evaluate the permeability of the sintered samples.

### 2.2. Gas Porosity and Permeability of Pre-Sintered Samples

An automated permeameter-porosimeter, Geologika PIK-PP (Russia) (Figure 4), was used for porosity and permeability measurements of pre-sintered samples. The device is a piece of standard petrophysical laboratory equipment for measuring medium-high gas porosity and absolute permeability values of porous cylindrical samples. The stack of pre-sintered ceramic samples was inserted into the measuring chamber; the air in the inlet and outlet chambers was evacuated before the testing. For measuring porosity, both the inlet and the outlet chambers were filled with helium. The device recorded dynamic pressure-volume characteristics of the gas and computed porosity, using Boyle-Marriott’s law. For measuring the permeability, the inlet chamber was filled with helium, and the gas flow was allowed across the sample. The device computes apparent permeability values via recording pressure evolution in the chambers and numerically processing the data.

### 2.3. Low Permeability Measurements of Sintered Samples

A method and test equipment have been previously developed and assembled at the Institute of Drilling and Fluid Mining of TU BAF to measure the permeability of cap rocks and salt rocks, in order to ensure their tightness and applicability for underground storage [31,32,33].

A unique low permeability measuring setup, developed at TU BAF, was employed in this study for transient measurements of permeability values. The stationary measurements that are especially useful for high permeability sample applications require the principle pressures and flow rates to be temporally constant, which implies the measurable flow of fluid through the sample. If the amount of fluid flowing through the sample is so small that it cannot be detected within acceptable measurement time, the transient measurement method (the pressures and flow rates change in time) is used to achieve the required result [34,35,36].

A transient two-chamber method designed for cylindrical samples and applied in this study can determine porosities and permeabilities of down to 10^−24^ m^2^. The technique can examine ceramic materials, tight reservoir materials, salt rocks, granite, claystone, and building materials. For the transient measurement method, evaluation software adapting to continually changing requirements [33] has been previously developed at TU BAF and employed in this study.

The experimental setup (Figure 5) consists of two stainless steel pressure chambers for the test gas (*V_inlet_* and *V_Outlet_*) and one core holder for the test samples. The maximum operating pressure is up to 200 bar. The volume of the pressure chambers and the pressure lines is adjustable to 10–165 mL. Pressure transducers at the measuring chambers could be selected based on the target permeability range, to provide the maximum measurement error of less than 0.1%. Besides, to confirm the tightness of the setup components, reference tests are routinely performed using dummy samples.

The inlet chamber (*V_inlet_*) was charged with a gas of absolute pressure; in the outlet chamber (*V_Outlet_*), a lower pressure was set. Opening a connection between two pressure chambers over the sample starts the measurement. Data recording includes the pressure profiles at the input and output of the chambers. A numerical method for determining the porosity and permeability of the samples uses the recorded pressure evolution curves and the known sample geometry. A finite-difference model, coupled with an automatic parameter identification method [34,35,36], was applied for the numerical processing of the results.

Analysis of fluid dynamics forms the theoretical basis of numerical processing of the transient testing results (Figure 5). The unsteady state (time-dependent) flow regime in the pore space was described using the partial differential Equation (1) for real gas:(1)$$div\left(\;\frac{k\;p}{\mu \;z_{g}}\;grad\;p\right)=\frac{\varphi \;c\;p}{z_{g}}\;\frac{\partial p}{\partial t},$$
herep—fluid pressure;k—sample permeability;μ—fluid viscosity;$z_{g}$—real gas factor;φ—sample porosity;c—isotherm compressibility.

The chambers at the inlet and outlet of the sample contain a finite volume of fluid (including line feeds to the sample). The boundary conditions in case of a 1D model for the chambers are the following:(2)$${\left(-\frac{\rho kA}{\mu }\;\frac{\partial p}{\partial x}=\;V_{K}\rho \;c_{K}\;\frac{dp}{dt}\right)}_{\left(x=0\;\mathrm{or}\;x=L\right)},$$
whereA—sample cross-sectional area;ρ—fluid density;$V_{K}$—chamber volume;$c_{K}$—chamber compressibility.

The boundary conditions in Equation (2) physically mean that the mass influx of effluent to/from the sample (left side of the equation) is equal to the temporal change in mass in the chamber (the right side of the equation). At the start of measurement (initial condition), the pressure in the sample and the output chamber was equal to $p_{A}$, the pressure in the inlet chamber was $p_{E}$; as a rule, $p_{E}>\;p_{A}$.

This mathematical problem was solved using an infinite series approach. An approximate solution for the case of a boundary condition of the first kind was derived at the end of the core sample (constant pressure $p_{E}$ or infinitely large chamber volume $V_{Inlet}$), and constant values were given for compressibility and viscosity, as well as for small porosities. Zisser and Nover [34] also used the approximate solution for the study of sandstones:(3)$$p_{A}\left(t\right)=p_{E}+\left(p_{A0}-p_{E}\right)\times exp\left\{-\frac{k\;A\;t}{L\;\mu \;c_{K}V_{A}}\right\},$$
where$V_{A}$—output chamber volume;$p_{E}$, $p_{A0}$—constant pressure values in the inlet chamber and the starting chamber (at t=0);$p_{A}\left(t\right)$—time-dependent pressure at the outlet chamber.

A regression in time-permeability domain delivered this solution:(4)$$ln\frac{p_{A}\left(t\right)-p_{E}}{p_{A0}-p_{E}}=f\left(t\right),$$

The laboratory procedures require only the spatial one-dimensional (1D) solution of the flow equation (in X-direction). In this method, a volume flow measurement is not necessary—the time-dependent pressure curves deliver permeability and porosity of the sample by the approximate inverse solution of the problem in the form of a goal function minimization. The goal function used was the mean square error of the measured and calculated pressure values:(5)$$f=\sqrt{\frac{J}{N-1}}\to Min.\;,\;\;J=\sum_{i=1}^{i=N}\left\{w_{i}{\left(p_{i}^{calculated}-p_{i}^{measured}\right)}^{2}\right\},$$
where***N***—the number of measured values;***w***_i_—weighting factors (the sum of all weighting factors has to be 1).

The minimization of the goal function (5) was carried out with the optimization method of Gauss–Newton and Levenberg–Marquardt [37]. The minimization of the error f represents the necessary condition for an accurate solution to the inverse problem. However, it is not sufficiently concerning the uniqueness of the calibrated permeability and porosity values. In laboratory tests, the uniqueness typically requires measurements of clear pressure response in both chambers. Therefore, the measures deliver both the permeability and the actual effective porosity.

The permeability measurement using gas is technically more straightforward than using liquid because the saturation of the sample can be carried out by vacuum drying, and the compressibility of the chambers is determined solely by the chamber pressure. The compressibility of the chamber is made up of the two parts, as shown in Equation (6):(6)$$c_{K}=c_{V}+c_{g},\;c_{g}=\frac{1}{p}-\frac{1}{z_{g}}\frac{\partial z_{g}}{\partial p}{|}_{T},$$
where$c_{V}$—compressibility of the chamber volume;$c_{g\;}$—compressibility of the gas.

At the initial state of the system, atmospheric pressure was applied at the outlet chamber and in the pore space of the tested sample, while the inlet chamber had a pressure $p_{E}$. The equilibration process involved digital recording and monitoring of the two converging pressures at inlet and outlet. The numerical model dynamically (in real-time) recomputes the error of the goal function and determines the permeability and the porosity of the sample whenever the new data read from the experimental setup. A finite-difference model was, therefore, coupled with an automatic parameter identification method.

A hardware implementation of the described method is a stand-alone unsteady state apparatus (Figure 6). The measuring fluids include N_2_, H_2_, He, CH_4_, and liquids.

The measurements of porosity of sintered samples, which contributes to Equation (1), was conducted via estimating the outer dimensions of the samples with a caliper, weighing the sample with analytical scales, and using Archimedes law. The original Al_2_O_3_ powder, provided within the ceramic paste, had α-phase, which did not change due to sintering. Thus, the maximum theoretical density of the samples employed for porosity computations was 3.99 g/cm^3^.

### 2.4. Microstructural Characterization

The microstructural characterization of the samples was made with electron microscopy imaging using Versa 3D (Thermo Fisher Scientific, Waltham, MA, USA). The pre-sintered sample surface allowed the evaluation of the particle size distribution in the ceramic feedstock. Focused Ion Beam Scanning Electron Microscopy (FIB-SEM) method was applied to selected samples (not included to the stacks, since the method is destructive) to assess three-dimensional (3D) distribution of pore space in the sample volume. The sintered sample imaging also included the SEM and FIB-SEM procedures. The processing of the imaging results was carried out on with Avizo (Thermo Fisher Scientific, Waltham, MA, USA) software.

For estimation of the particle size, an SEM image was processed with a series of image-processing filters and segmented into many separate particles/labels. The labels were analyzed with Avizo’s internal “Label-Analysis” module to provide over 20 geometrical and morphological characteristics of each label. Based on the particle shape and morphology, the “Equivalent Diameter” characteristic was selected to represent the particle size measurement. The equivalent diameter represents the diameter of such a circle that has the area equivalent to that of the analyzed particle. This characteristic provides a realistic size measure for particles having aspect ratio near “1”—it is an acceptable estimate of the average particle size. A histogram plotted the computed equal diameter values.

## 3. Results

### 3.1. Microstructural Description

The cylindrical ceramic samples 3D-printed with SLA technology (Figure 7) were debinded and pre-sintered, then tested with electronic microscopy methods. Since particle size significantly affects the material performance during sintering, the microstructural characteristics of the samples were assessed with SEM, and the particle size-distribution histogram was built.

SEM imaging was used as a semi-analytical method to measure particle size via the image processing approach (Figure 8). The original SEM image was collected using a secondary electron detector over an area of over 0.3 × 0.2 mm (9619 × 6468 px) with pixel size 32 nm/px.

For the computation of particle size with image processing methods, the image was cropped to the size of 1900 × 2000 px (Figure 8a). A few image processing filters (such as non-local means, unsharp masking, and others) were applied to the grayscale image to flatten the intensity gradients in the inner regions of the particles and to increase the contrast at the particles’ boundary. The h-maxima watershed method was applied to the processed image to get watershed basins, corresponding to the particles. A unique label within a global label-field represented each particle (Figure 8b). The label-analysis module was applied to the label field to compute Feret measures, area, perimeter, and other characteristics. As a derivative of the area measure, the equivalent diameter was calculated and plotted in the form of a histogram (Figure 8c). The histogram indicates that the particle size of the alumina powder mixed in the paste is in the range of 0.2–3 μm. Analysis of the original image revealed that single particles of 5–10 μm occur sporadically over the examined area.

The results included separate high-resolution SEM images for the debinded/pre-sintered samples (Figure 9a) and the sintered samples (Figure 9b). The former samples have the highest porosity achievable for the investigated material via SLA manufacturing procedure. The images show loosely consolidated particles in Figure 9a, with minor bridging occurring at the contact boundaries between the particles. Such material topology leads to the formation of multiple porous interconnected passages that determine the high permeability of the pre-sintered material. For the sintered particles (Figure 9b), the bridging is well developed. The large particles maintain their shape but have softened contours at the boundaries. Most of the small particles merged with the large particles and served as the primary material for bridging via surface diffusion mechanisms.

Besides the SEM imaging, there were FIB-SEM 3-dimensional models acquired from the debinded/pre-sintered and sintered samples. The stack of the original FIB-SEM images (Figure 10) was aligned and processed with image-processing filters. Segmentation of the grayscale stack led to the 3D representation of the pore-space for pre-sintered and sintered ceramic samples. The quality of the grayscale images for FIB-SEM models is lower than the dedicated SEM images (Figure 9) because a stack of images was collected while removing the material layer-by-layer with an ion beam. Both, the ion beam action and the use of an electron beam for imaging promote surface charging, which is especially critical for Al_2_O_3_ material. The surface charges reduce the imaging quality of an electron microscope.

The pore space of the debinded/pre-sintered samples has a well-developed network of pores and channels that interconnect each other and form a complex-shaped structure. The complexity of the pore-space structure is due to multiple ceramic particles of different sizes occupying the analysed volume. In contrast, a significant fraction of pore space of the sintered sample includes large ceramic grains that consist of merged small and medium-sized particles with softened outer boundaries. The pores are formed only in the corners between neighbouring grains, which is typical for ceramics sintered in a gaseous environment. Some pores in the sintered samples are closed, while others participate in the formation of the open porosity.

Based on FIB-SEM 3D models (Figure 10a) of the debinded/pre-sintered samples, the multiple thin interconnected channels form fluid filtration pathways. This topology determines the large permeability values of the debinded/pre-sintered samples. The filtration pathways of the sintered samples (Figure 10b) primarily consist of relatively large pores connected with thin nanochannels. Such a pore network allows the filtration of gas and hinders the filtration of liquids through the nanochannels. It also severely limits the permeability values of the sample.

### 3.2. Gas Porosity and Permeability

The stack of debinded/pre-sintered cylindrical samples was characterized in terms of gas porosity and permeability (Table 1). A composite cylindrical sample consisted of six ceramic discs Ø30 × 3–5 mm. A single measurement took at average 15 min. Repeatability study included five subsequent analyses without unloading the sample from a core holder. Gas open porosity averaged to 36%, while gas (absolute) permeability—to 2.37 × 10^−15^ m^2^. Slight permeability trend may indicate degradation of the sample during subsequent measurements associated with mechanical destruction/disintegration at each confining pressure release step.

Quality control tests included multiple measurements on a commercial set of ceramic reference standard cylinders with dimensions Ø30 × 30 mm. Metrological tests showed that random error of a single pore volume measurement (excluding the first measurement) did not exceed ±0.02 cm^3^. Permeability measurements during the tests showed precision better than ±1%.

### 3.3. Low-Permeability Measurements

The sintered samples were assembled into stacks of 30 mm thick and used for the measurements. A rubber gasket isolated the side face of the stack from a spurious crossflow. The preliminary measurements indicated that the testing time for the stacks was extremely long. Therefore single 5 mm thick cylinders, instead of stacks, participated in further testing. The testing of permeability values for a single 5-mm thick sintered cylinder took up to 11 days (Figure 11). The porosity values for the tested samples were in the range of 6.1–6.5%.

The pressure profiles at the input and output exhibit changes in behavior not earlier than after 100,000 s (ca. 28 h) of measurements, the total length of measurements was over 11 days for each low-permeable sample. The numerical model was recomputed continuously, starting from the beginning of the test until the end. The error evolution of the numerical approximation was tracked using a discrepancy between measured and simulated permeability values. When the difference between the measured and calculated pressure values became smaller than 10%, the test was stopped, and permeability value was estimated as 1.8 × 10^−21^ m^2^ for one sample (Figure 11a) and 3.7 × 10^−19^ m^2^ for another sample (Figure 11b). Figure 11c shows the pressure evolution profile obtained from a ceramic sample of higher permeability of 3.7 × 10^−15^ m^2^—it demonstrates the typical shape of the complete pressure profile, as well as indicates that measuring time for materials with medium permeability is orders of magnitude shorter than that for low-permeable ones. 

The reproducibility of the results was verified via repetitive testing of the same sample (Figure 12). After the first run (Figure 12a), the measured initial permeability value was 2.0 × 10^−20^ m^2^. After four more runs, in the fifth run (Figure 12b), the measured permeability value was 1.5 × 10^−20^ m^2^. The obtained results indicate that the reproducibility is ensured.

## 4. Discussion

The volume fraction of the ceramic powder present in the paste affects particle packing efficiency during 3D printing, influences surface and bulk diffusion processes of ceramic molecules during inter-particle bridge forming in the sintering process and determines the seeds for grain generation during sintering. Since pores and channels of the sintered materials are aligned to the corners and edges of the ceramic grains, the size of the grains regulates the potential pore-space characteristics of the sintered ceramic material. A combination of powders of different fractions could be used for tuning the pore-space of the sintered material.

Another factor related to powder particles is their morphology. The Al_2_O_3_ powder used in this study had particles with sizes distributed in a broad range of values. The particles have irregular flat shape, which makes it challenging to develop a predictive model of pore-space evolution during sintering. At the same time, there are other powders available that consist of uniform spherical particles and could be used to improve the predictability and repeatability of the result. 

During sintering, the significant factors affecting material compaction are the absolute temperature at each stage of heating, the heating/cooling rate, which is known to affect the grain growth intensity, and the dwell values at each particular temperature. Altogether they constitute the sintering curve. Multiple studies appeared in the past decades for traditionally manufactured ceramics in an attempt to characterize the evolution of the relative density of the material [38,39,40,41]. However, there are no comprehensive data available on the evolution of material microstructure, and especially there are no data on the evolution of material permeability. Thus, despite a large amount of the available scientific data on ceramics behavior, to the moment, there could be no connection established between the process parameters of ceramics manufacturing and the resultant pore-space characteristics of the material. 

The 3D printed ceramics features anisotropy in shrinkage during sintering, which is attributed to the in-layer and across-layer directions of the printing process. The anisotropy often reaches a 5–10% difference in linear shrinkage in different directions. This factor differentiates the 3D printed ceramics from traditional manufactured ones and must be accounted for when considering pore-space characteristics of additively manufactured items. There are almost no data available for additively manufactured ceramic materials, even to describe the sintering behavior of particular types of ceramics.

The authors [42] investigated porosity evolution as a function of sintering temperature and developed a preliminary predictive numerical model for 3D printed ceramic shrinkage. For the porosity range investigated in [42], there were two terminal values of permeability determined in this study (schematically shown with red circles in Figure 13), which determine the minimal and maximal values of permeability achievable with the studied material. The porosity curve in Figure 13 does not account for dwell values, while the shown permeability values do. It is imperative to restore the shape of the permeability evolution curve (shown with dashed circles in Figure 13), to better understand the evolution of fluid transport properties of the sintered ceramics. While the left part of the shown permeability range could be assessed with regular laboratory equipment, after a particular value of sintering temperature, the permeability becomes so small, that special equipment, designed for low-permeability measurements, needs to be involved.

The target applications of low-permeable materials often must deal with the prevention of fluid diffusion through the low-permeable media. The accurate measurement of such tight materials is challenging because the fluid flow rates in the test have to be on par with those of diffusion processes. Therefore, potential underground storage projects require permeability of 10^−21^ to 10^−22^ m^2^. Consequently, any measurement method applied to determine such low permeability should have a detection threshold of 10^−23^ m^2^ or even less. 

For sintered ceramics, the porosity is a better controllable parameter, than permeability [43]. However, the permeability of sintered ceramics is a crucial parameter for many practical applications. Therefore, the development of a robust and time-efficient workflow for measuring low porosity and permeability of the ceramic samples seems to be a significant contribution to the technological process. It requires a further improvement of the numerical interpretation model for the low permeability testing method. The results show that a full measurement cycle on a sintered ceramic sample takes more than ten days that yield two serial measurements a month. Thus, one laboratory setup can measure less than 24 samples a year in the best-case scenario. Such a low number of samples would substantially limit the practical applicability of the approach. In this context, the current data interpretation model needs to be enhanced in the way of obtaining reliable estimates of low permeability within a much shorter time.

Another essential consideration is permeability testing for complex 3D sample geometries, instead of simple primitives such as cylinders. The simple shape of the 3D-printed ceramic samples allows both direct evaluation of 3D printing and sintering [44], as well as the low permeability testing, as we show here. The coupling enables sustaining homogeneity of void space structure within the sample volume. However, in the case of 3D printing and sintering of complex geometries, the resulting samples might possess substantial volumetric heterogeneity. Therefore, some regions of the sintered sample may have altered permeability. Eliminating this effect may require not only the development of a novel method for low permeability testing of non-primitive shapes but also further research on predictive simulation of printing and sintering of samples having a complex shape.

## 5. Conclusions

This study pursued two objectives. First, the transport properties of 3D printed ceramics were assessed concerning the maximal and minimal values that could be achieved via manipulation of the manufacturing process parameters. Second, a unique laboratory setup developed for measuring low permeability values was tested with ceramic samples and provided accurate, repeatable results. The combination of these methods provides a technological opportunity to engineer and manufacture porous and permeable materials with controlled storage capacity (porosity), transport properties (permeability), and complex 3D shape. One of the attractive application areas for such samples is the creation of reference permeability standards.

The presented study shows that a commercially available 3D printing SLA technique allows manufacturing samples with open porosity in a range of 5%–36% and absolute permeability from 1.8 × 10^−21^ m^2^ to 3 × 10^−15^ m^2^, correspondingly. A standard sintering technique was used to modify the internal structure of ceramic material and reduce its permeability. The results of the low permeability test indicate that the employed experimental setup successfully measures the permeability values of 3D printed low-permeable ceramics. Results reproducibility was confirmed. The results of the study provide an opportunity for further development of 3D printed materials with controlled porosity and permeability, with the potential to produce nano-permeable materials, suitable for the use as reference permeability standards. 

## Figures and Tables

**Figure 1 materials-12-03886-f001:**
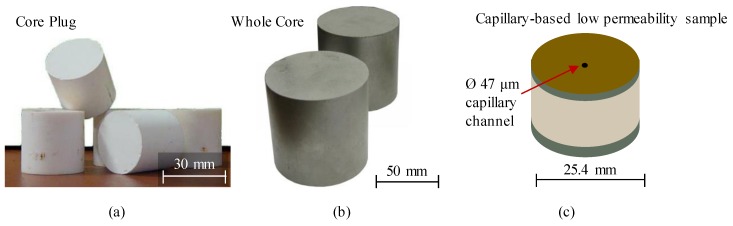
Illustration of typical reference porosity-permeability standard samples (**a**) and (**b**) for medium-to-high permeability range (courtesy of EcoGeosProm LLC), (**c**) for low permeability range, similar to [28].

**Figure 2 materials-12-03886-f002:**
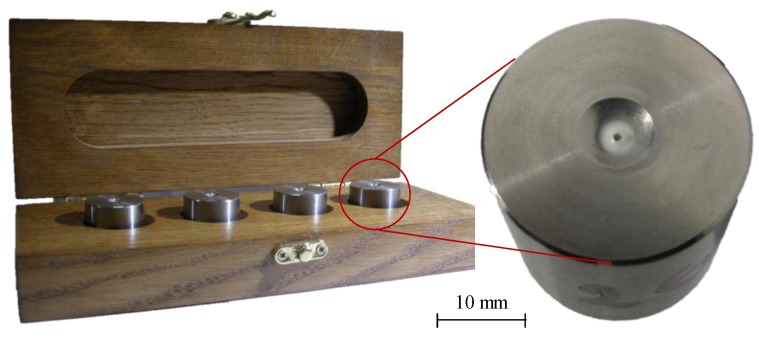
A commercially available kit of Ø30 mm capillary-based permeability standards for calibrating laboratory nano-permeameters (courtesy of I-Texx Engineering).

**Figure 3 materials-12-03886-f003:**
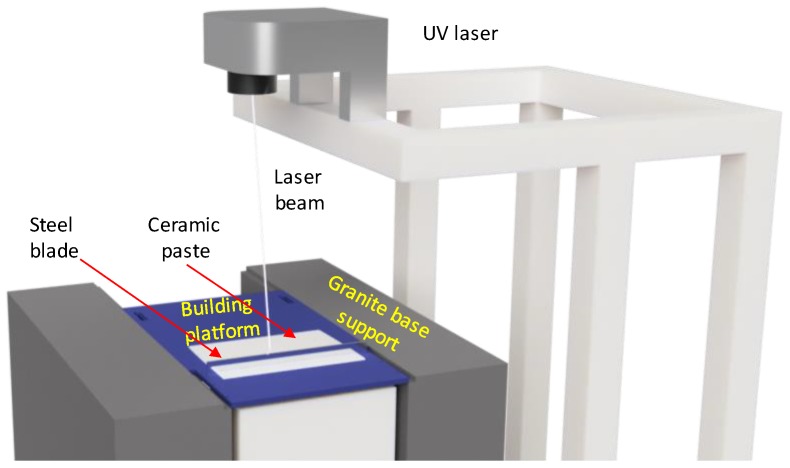
Schematic diagram of ceramic 3D printing with a Ceramaker 900 unit.

**Figure 4 materials-12-03886-f004:**
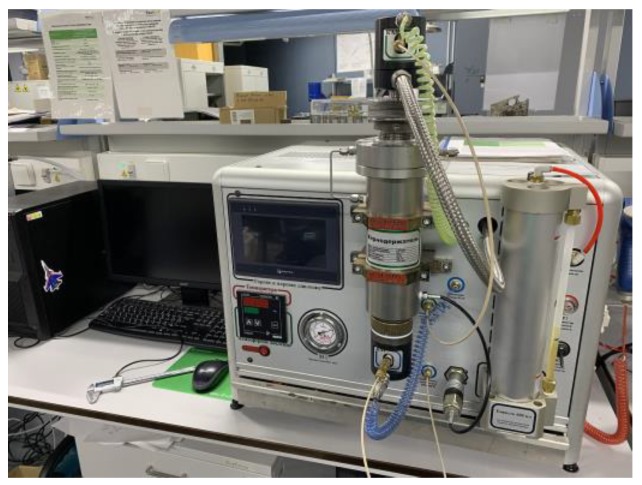
An automated permeameter-porosimeter Geologika PIK-PP (courtesy of Skoltech).

**Figure 5 materials-12-03886-f005:**
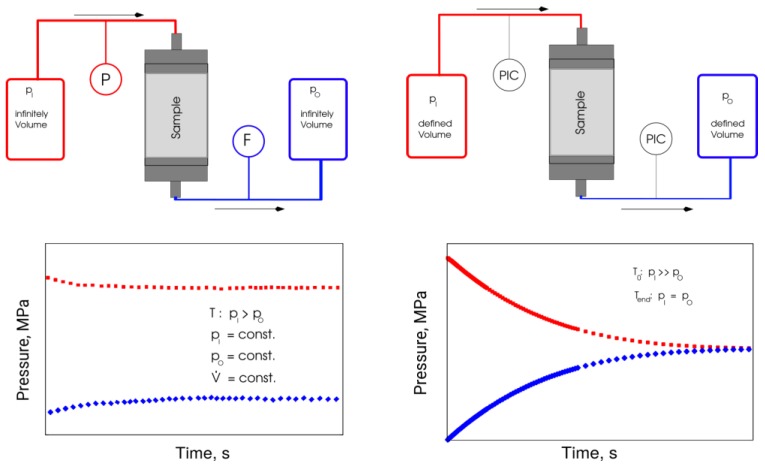
Principle measuring procedure of stationary (left) and transient (right) methods.

**Figure 6 materials-12-03886-f006:**
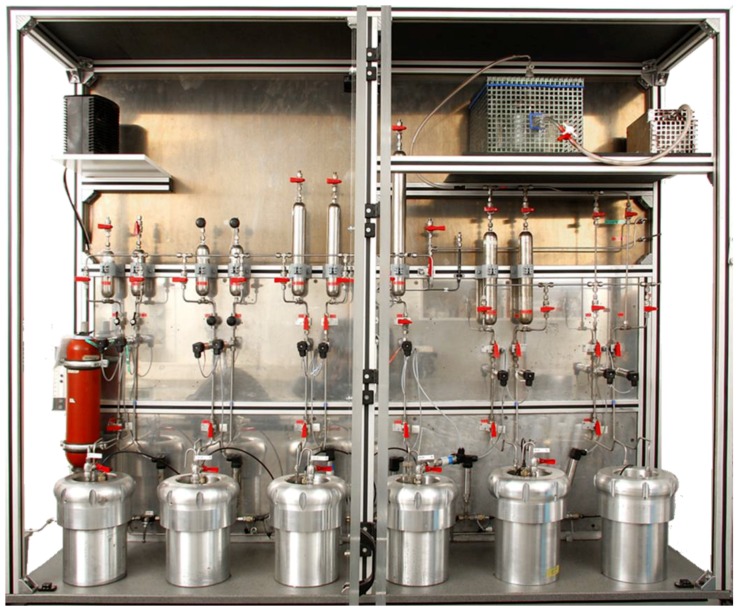
Unsteady state apparatus for low permeability measurements, presented in detail in [31].

**Figure 7 materials-12-03886-f007:**
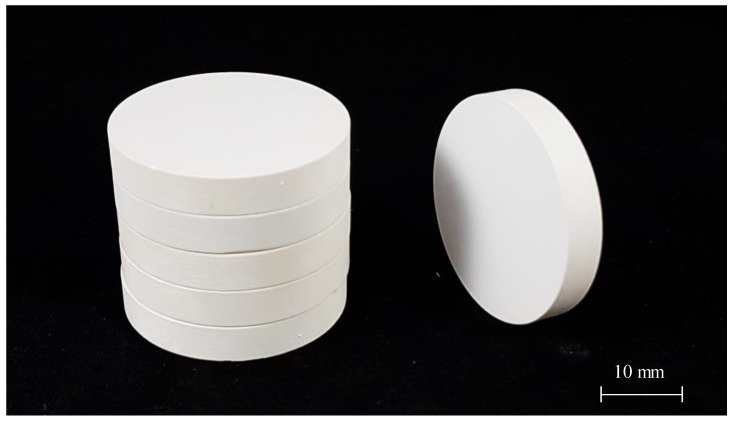
Al_2_O_3_ cylindrical samples 3D-printed with stereolithography (SLA) technology for permeability testing.

**Figure 8 materials-12-03886-f008:**
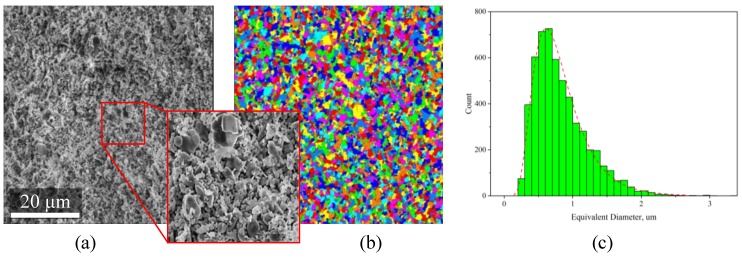
Estimation of particle size distribution via image processing of large-area SEM images of pre-sintered ceramic samples: (**a**) original SEM image, (**b**) segmented image, (**c**) particle size distribution.

**Figure 9 materials-12-03886-f009:**
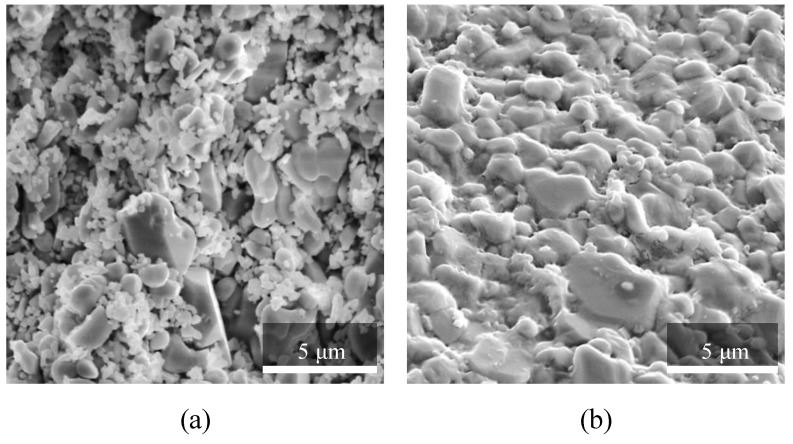
SEM images of a broken surface of Al_2_O_3_ samples, manufactured with SLA technology: (**a**) debinded/pre-sintered and (**b**) sintered.

**Figure 10 materials-12-03886-f010:**
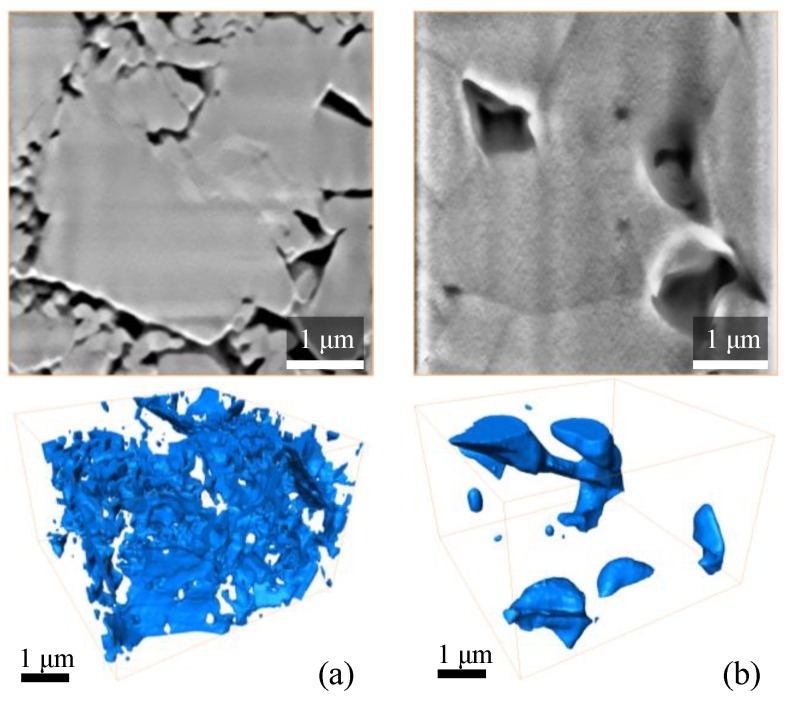
Typical FIB/SEM slices and 3D reconstructed model of pore-space of Al_2_O_3_ samples, manufactured with the SLA technology: (**a**) debinded/pre-sintered and (**b**) sintered.

**Figure 11 materials-12-03886-f011:**
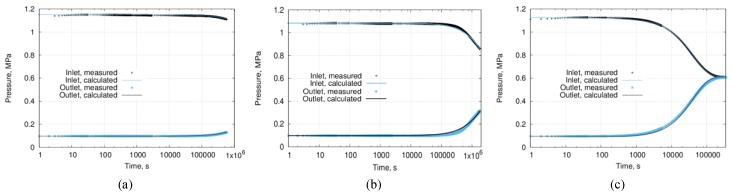
Transient pressure profiles of three sintered ceramic samples; (**a**) a sintered sample’s estimated permeability is 1.8 × 10^−21^ m^2^, (**b**) a sintered sample’s estimated permeability is 3.7 × 10^−19^ m^2^, and (**c**) a pre-sintered sample’s (complementary to the tested stack) estimated permeability is 3.7 × 10^−15^ m^2^.

**Figure 12 materials-12-03886-f012:**
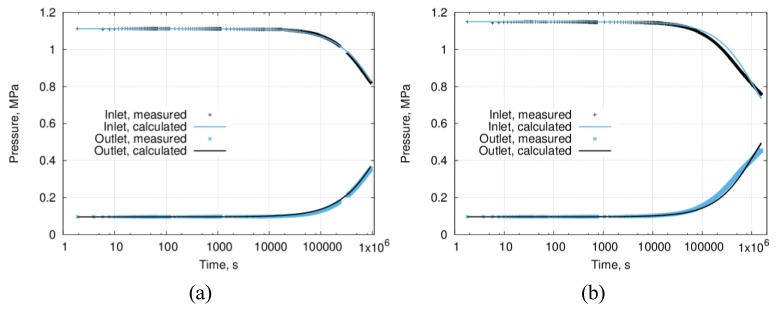
Reproducibility of the low permeability measurements for sintered ceramic samples, consequently conducted with the same sample: (**a**) the first run—permeability 2.0 × 10^−20^ m^2^ and (**b**) the fifth run—permeability 1.5 × 10^−20^ m^2^.

**Figure 13 materials-12-03886-f013:**
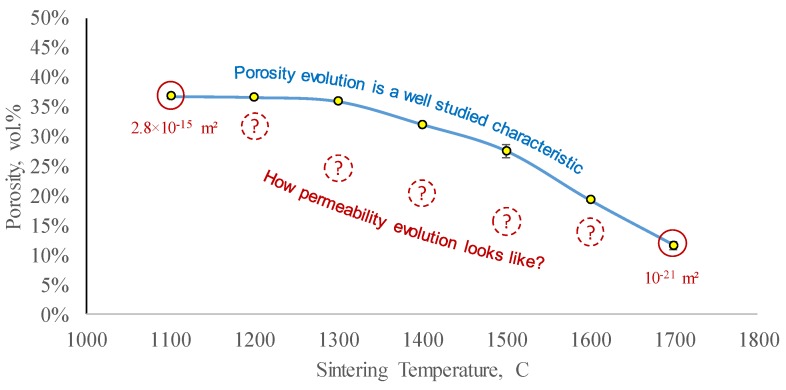
The evolution of porosity during Al_2_O_3_ ceramic sintering is a typical characteristic to control. The values shown in solid red are the terminal values of permeability measured in this study—to identify the available range. A complete curve (schematically shown with dashed red circles) is required to accurately model permeability evolution during sintering (the porosity data were originally presented in [42]).

**Table 1 materials-12-03886-t001:** Summary of gas porosity and permeability measurements.

Replicate	Sample Length (cm)	Sample OD (cm)	Sample Cross-Sectional Area (cm^2^)	Sample Volume (cm^3^)	Confining Pressure (MPa)	Pore Pressure (MPa)	Gas Porosity (%)	Gas Permeability × 10^−15^ (m^2^)	Klinkenberg-Corrected Gas Permeability × 10^−15^ (m^2^)
1	↑ 3.058 ↓	↑ 3.007 ↓	↑ 7.1016 ↓	↑ 21.72 ↓	↑ 3.44738 ↓	7.85	36.0	2.69	2.29
2						7.81	36.0	2.76	2.35
3						7.81	36.0	2.78	2.37
4						7.81	36.0	2.81	2.40
5						7.82	36.0	2.85	2.43
						**Average:**	**36.0**	**2.78**	**2.37**

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
