# Peer review of "Towards Creation of Ceramic-Based Low Permeability Reference Standards"

_materials, 2019, doi:10.3390/ma12233886_

Round 1

Reviewer 1 Report

This is a very good paper that introduced a new process for fabricating a low-perm standard rock that can be used in experimental calibration. I am very interested in this topic and it is beneficial to read it.

Reviewer 2 Report

see attached

Reviewer 3 Report

The article is devoted to the study of ceramic materials with controlled porosity and permeability. Due to the constantly growing range of potential directions of application of this type materials, the article seems to be a valuable contribution to the state of knowledge about the properties of these materials. The range of tested series of samples, although modest, allowed to achieve the intended research objectives. However, it seems that a valuable supplement would be to broaden the discussion by comparing the results of porosity and permeability to other mineral materials, such as other ceramic materials or advanced cementitious composites.

Detail remarks: 
1. line 205; 218; 239; 248; 350 "Error! Reference source not found" should be corrected,

2. complete the unit in the y-axis description in figure 8 (c),

3. line 357 - Table 1 – it seems to be surprising five identical gas porosity results while the material tested show varying gas permeability

4. line 350 - please explain why the sample used in permeability tests consisted of six ceramic discs of varying heights. Is it possible to conduct tests for individual ceramic discs as in the case of Low-Permeability Measurements tests?

5. line 414-415 "...Multiple studies appeared in the past decades for traditionally manufactured ceramics in an attempt to characterize the evolution of the relative density of the material..." - please add references,
